# Assessment on Different Vaccine Formulation Parameters in the Protection against Heterologous Challenge with FMDV in Cattle

**DOI:** 10.3390/v14081781

**Published:** 2022-08-15

**Authors:** Sebastián Di Giacomo, Danilo Bucafusco, Juan Manuel Schammas, Juan Pega, María Cruz Miraglia, Florencia Barrionuevo, Alejandra Victoria Capozzo, Daniel Mariano Perez-Filgueira

**Affiliations:** Instituto de Virología e Innovaciones Tecnológicas, Centro de Investigaciones en Ciencias Veterinarias y Agronómicas (CICVyA), Instituto Nacional de Tecnología Agropecuaria-Consejo Nacional de Investigaciones Técnicas (INTA-CONICET), N. Repetto y De Los Reseros s/n (1686), Hurlingham 1686, Buenos Aires, Argentina

**Keywords:** FMDV, cattle, heterologous protection

## Abstract

Foot-and-mouth disease (FMD) remains one of the major threats to animal health worldwide. Its causative agent, the FMD virus (FMDV), affects cloven-hoofed animals, including farm animals and wildlife species, inflicting severe damage to the international trade and livestock industry. FMDV antigenic variability remains one of the biggest challenges for vaccine-based control strategies. The current study analyzed the host’s adaptive immune responses in cattle immunized with different vaccine protocols and investigated its associations with the clinical outcome after infection with a heterologous strain of FMDV. The results showed that antigenic payload, multivalency, and revaccination may impact on the clinical outcome after heterologous challenge with FMDV. Protection from the experimental infection was related to qualitative traits of the elicited antibodies, such as avidity, IgG isotype composition, and specificity diversity, modulating and reflecting the vaccine-induced maturation of the humoral response. The correlation analyses of the serum avidity obtained per vaccinated individual might suggest that conventional vaccination can induce high-affinity immunoglobulins against conserved epitopes even within different FMDV serotypes. Cross-reaction among strains by these high-affinity antibodies may support further protection against a heterologous infection with FMDV.

## 1. Introduction

Foot-and-mouth disease (FMD) is listed by the World Organisation for Animal Health (WOAH) as a notifiable disease, owing to its potential for rapid spread and severe economic consequences, irrespective of national borders. The WOAH considers the FMD virus (FMDV) the most contagious infectious agent in mammals, and FMD was the first official WOAH notifiable disease [1]. Its etiological agent is a small non-enveloped positive-stranded RNA virus, the sole member of the Aphthovirus genus within the Picornaviridae family [2]. FMDV affects all domestic biungulates; wildlife species may act as reservoirs under certain ecological conditions [3]. Lethality has been described only for young animals and certain FMDV strains [4]. However, its main disruptive potential is the high morbidity rate and the indirect losses associated with outbreaks in territories with susceptible populations. FMD incursions may result in severe and far-reaching economic losses, interrupting regional and international trade in developed countries [5,6], and affecting production efficiency and genetic diversity in developing regions due to the loss of animals [7].

The virulence, wide range of hosts, diversity of variants, and high infectious and contagious capacity of the FMDV explain its presence and constant reemergence worldwide and identify FMD as a health problem on a global scale [8]. FMD is endemic in several regions of Africa and Asia. Most areas in South America are FMD-free, presenting zones under vaccination programs and others without vaccination [9]. However, the reintroduction of the disease detected in countries, such as Paraguay [10], Ecuador [11], and Colombia [12], previously declared free of FMD with vaccination by the WOAH, reinforces its current relevance.

Conventional FMD vaccines comprise chemically inactivated whole viral particles formulated in oil or aqueous adjuvants [13]. Good quality commercial vaccines may prevent clinical FMD and transmission to other susceptible animals after infection with homologous viral strains [14]. Some of the immune mechanisms involved in protection against the homologous virus challenge have been described previously [15,16]. Still, the protective ability of any vaccine must also contemplate its capacity to respond against viral strains not present in the formulation [17]. This particular problem, closely associated to the phenotype of the FMDV and its hosts, remains one of the biggest challenges for FMD control worldwide [18].

This work analyzes some of the mechanisms and immune factors involved in protection against heterologous infection with FMDV in cattle. To this end, a series of in vivo heterologous challenge experiments were performed, using groups of cattle previously vaccinated with different single-oil emulsion conventional FMD formulations and immunization regimes. Our results demonstrate that antigenic payload, multivalency, and revaccination may impact the clinical outcome after challenge with a heterologous strain of FMDV. The study of the humoral responses elicited before experimental infection showed that protection to clinical FMD correlated with qualitative aspects of the adaptive humoral response, such as avidity and specificity of the antibodies induced after vaccination, rather than simply quantitative parameters, such as their total or neutralizing antibody titers.

## 2. Materials and Methods

### 2.1. Experimental Animals

Twenty-two FMD-unvaccinated calves (180–220 kg each, 6- to 8-months old) were purchased from a livestock breeder from Buenos Aires province (Tandil, Argentina), located within the FMDV-free region with vaccination. All animals were checked by liquid-phase blocking enzyme-linked immunosorbent assay (LPBE) for the absence of colostral FMDV-specific antibodies upon arrival at the field of the Centro de Investigaciones en Ciencias Veterinarias y Agronómicas, Instituto Nacional de Tecnología Agropecuaria (CICVyA-INTA). After experimental vaccination, animals were kept in the CICVyA-INTA field until performing the virus challenge with virulent FMDV. Experimental infections were performed at the BSL-4 WOAH animal box facilities located at the CICVyA-INTA. All assays were performed following biosafety and animal welfare regulations, according to protocol 52/2013 approved by the Institutional Committee for Use and Care of Experimental Animals (CICUAE) from the CICVyA-INTA. 

### 2.2. Experimental Design and Sampling

Three experimental single-oil emulsion FMD vaccines, manufactured by Biogenesis-Bagó S.A., controlled and approved by the National Agrifood Health and Quality Service (SENASA), were used in the experiments: two monovalent formulations containing 10 µg or 40 µg per dose of inactivated FMDV A24/Cruzeiro/Brazil/55 (A24/Cruzeiro) and a trivalent formulation containing inactivated FMDV strains A24/Cruzeiro (10 µg/dose), O1/Campos/Brazil/58 (O1/Campos, 20 µg/dose), and C3/Indaial/Brazil/71 (C3/Indaial, 10 µg/dose). Steers were randomly distributed into four experimental groups (*n* = 5 each), and vaccines were applied subcutaneously in the neck (2 mL/dose), following the immunization protocols assigned to each experimental group (Table 1). Animals from the revaccinated group (*A24 10 µg × 2*) received a second immunization at 15 days post-primary vaccination (dpv). Two animals were kept as naïve (non-vaccinated) controls. Serum, plasma, and whole blood samples were obtained from the jugular vein using Vacutainer^®^ (BD, Franklin Lakes, NJ, USA) tubes at 0 and 30 dpv.

### 2.3. Experimental Infections and Clinical Assessment of Cattle

Virulent FMDV strain A/Argentina/2001 (A/Arg/01) was provided by the WOAH FMD Reference Laboratory at SENASA, Argentina. Challenge was performed at 30 dpv by inoculation of 10,000 tissue culture infectious doses 50% (TCID_50%_) through the intradermolingual route. After the challenge, cattle were monitored daily for clinical signs of FMD. These included vesiculation, lameness, increased salivation, loss of appetite, and fever (rectal temperature of >39.5 °C). Clinical disease progression was determined by assigning scores of 1 for fever > 40.0 °C, 1 for lesions in the oral and nasal cavities (except the inoculation site in the tongue), 1 for each foot that developed vesicles, and 0.5 for fever between 39.5 °C and 40 °C, with a maximum daily clinical score of 5. Animals with clinical scores < 1 were considered “protected”.

### 2.4. Inactivated FMDV Antigens

Inactivated and concentrated preparations of FMDV strains A24/Cruzeiro, O1/Campos, A/Arg/01 and C3/Indaial were kindly provided by Biogenesis-Bagó S.A. Whole FMDV purified particles used for serology and cellular immunity assays were obtained by purification from these inactivated preparations, following a 15–45% sucrose density gradient centrifugation method [19], further optimized in our laboratory [20].

### 2.5. Assessment of FMDV-Specific Total and Neutralizing Antibodies

Total antibodies against the four FMDV strains utilized in the experiments were measured by a liquid-phase blocking ELISA (LPBE) originally developed by Hamblin et al. [21] and further modified by Periolo et al. [22,23]. FMDV-neutralizing antibodies (Nab) against the A24/cruzeiro and A/Arg/01 strains were detected by a microplate virus neutralization test (VNT) as described in [24]. VNT titers were calculated as the log_10_ of the reciprocal serum dilution required for 50% neutralization of 100 TCID_50%_ of FMDV, according to the Reed and Muench method [25].

### 2.6. Indirect Reference Parameters for Assessment of the Heterologous Protection

Antibody titers induced after vaccination and measured by LPBE and VNT were also referred to the “expected percentage of protection” (EPP) already established for the A24/Cruzeiro and A/Arg/01 strains. The EPP estimates the likelihood that cattle would be protected after homologous FMDV challenge based on the specific antibody titers measured before challenge. EPP values for the A24/Cruzeiro and A/Arg/01 strains arise from correlations between the LPB-ELISA [23,26] or VNT [27,28] titers obtained in vaccinated cattle at 60 dpv, and the in vivo challenge results obtained at 90 dpv by the “Protection against Podal Generalization” (PPG) method, involving 16 vaccinated animals infected with the homologous strain. For both total and neutralizing FMDV-specific antibodies, the EPP ≥ 75% (EPP_75_) values serve as a reference of the antibody titers associated with the protection at population level against the homologous challenge with the A24/Cruzeiro or A/Arg/01 strains.

### 2.7. Isotype Profiles and Avidity of the FMDV-Specific Antibody Responses

Antibody isotype responses were measured against the A24/Cruzeiro and A/Arg/01 FMDV strains. FMDV-specific IgM antibodies were detected using a double sandwich ELISA (Di Giacomo et al. manuscript in preparation). Briefly, 96-well ELISA plates (MICROLON^®^, Geiner Bio-One, Kremsmunster, Austria) were coated ON at 37 °C with a sheep anti-bovine IgM serum (BioRAD, Hercules, CA, USA) diluted 1:1000 in carbonate-bicarbonate buffer 50 mM pH = 9.6. The next day, plates were incubated for 1 h at 37 °C in blocking buffer (BB), containing PBS 0.05% Tween 20 (PBST) 10% normal equine serum. Plates were washed in PBST and serial dilutions of bovine serum samples in BB were then incubated for 1 h at 37 °C. After PBST washing, inactivated purified A24/Cruzeiro or A/Arg/01 FMDV suspensions (20 ng/well) were added to the plates for 1h at 37 °C. Following PBST washing, plates were incubated for 1h at 37 °C with a FMDV-specific guinea pig hyperimmune sera against the corresponding viral strain. Reactions were finally revealed using anti-guinea pig IgG-HRP conjugate (KPL, Kidderminster, United Kingdom), followed by addition of o-phenylenediamine (OPD) peroxidase substrate (KPL, Kidderminster, UK) for 15 min and H_2_SO_4_ 2 M to stop color development. FMDV-specific bovine IgG isotypes were detected by ELISA, as reported by Lavoria et al. [10], except that HRP-conjugates anti-IgG1 and -IgG2 antibodies (AbD Serotec) were used diluted 1:750 in BB. For all assays, serum samples were run in two-fold serial dilutions starting at 1:50 and isotype antibody titers were expressed as the log_10_ of the highest dilution of the serum reaching an optical density (OD) equal to the mean OD obtained from eight negative bovine sera ± 2 standard deviations (SD).

Avidity of the anti-FMDV antibodies was determined for whole IgG, IgG1, and IgG2 immunoglobulins, following the same procedures described by Lavoria et al. [29], except that HRP-conjugate anti-bovine IgG (H + L, Jackson Laboratories, Philadelphia, Pennsylvania) was used diluted 1:5000 for total avidity assessment while HRP-conjugates anti-IgG1and -IgG2 antibodies (AbD Serotec) were used diluted 1:750. The avidity index (AI) was calculated as the percentage of residual activity of the serum sample after urea washing, relative to that of untreated sample: AI% = (OD sample with urea/OD sample without urea) × 100.

### 2.8. Evaluation of the FMDV-Specific Interferon (IFN)-γ Production

The induction of antigen-specific cellular immune responses was established through the in vitro production of interferon (IFN)-γ in whole blood samples taken from vaccinated or naïve animals at 30 dpv and stimulated with inactivated FMDV strains A24/Cruzeiro and A/Arg/01, as previously developed in our laboratory [30]. Briefly, whole blood samples (1.5 mL) were incubated in 24-well sterile cell culture plates (Nunc, Ocala, FL, USA) during 24 h at 37 °C/5% CO_2_ with PBS, Pokeweed mitogen (10 μg/mL), or purified inactivated FMDV A24/Cruzeiro or A/Arg/01 (10 μg/mL). Plates were centrifuged for plasma collection, and IFN-γ produced in each well was analyzed using a commercial ELISA (BOVIGAM^®^, Thermo Scientific, Waltham, MA, USA), performed according to the manufacturer’s specifications. IFN-γ levels were expressed as pg/mL of plasma using a standard curve build with known concentrations of a recombinant bovine IFN-γ (AbDSerotec, Oxford, UK) ranging from 195 to 25,000 pg/mL.

### 2.9. Assessment of Virus Replication

Viral replication was determined after FMDV challenge in naïve and vaccinated steers following serological and direct virus isolation methods. Indirect serological assessment consisted of the detection of antibodies against FMDV non-structural proteins (NSP) in serum samples obtained at 7 days post-infection (dpi), using a commercial ELISA kit (PrioCHECK^®^ FMDV NS, Thermo, Alpharetta, GA, USA), according to the manufacturer’s instructions. Direct virus isolation was performed on BHK-21 (C-13) cell monolayers (200 μL/sample) incubated with whole heparinized blood samples taken at 4 dpi from animals with clinical FMD and one protected steer per experimental group as negative controls. Samples were seeded in duplicate in 48-well cell culture plates (CELLSTAR^®^, Geiner Bio-One, Kremsmunster, Austria) and each trial included infective FMDV strain A/Arg/01 as a positive control and an autologous blood sample obtained before vaccination as negative control. After 1 h of incubation at 37 °C and 5% CO_2_, culture plates were washed four times with sterile PBS and further incubated adding 500 μL/well of D-MEM culture medium 2% of fetal calf serum (FBS). Plates were maintained for 72 h at 37 °C and 5% CO_2_ and examined for the development of cytopathic effect (CPE) compatible with the presence of virulent FMDV in the sample. In those wells where no CPE was observed after 72 h, both the monolayer and supernatant were harvested and frozen at −20 °C for 16 h. Next, they were thawed and clarified by centrifugation at 1200× *g* for 5 min and incubated once again on BHK-21 monolayers, as described above. These “blind passages” were repeated up to three times for CPE-negative blood samples to be considered as “negative” for virus isolation.

### 2.10. Statistical Analysis

Comparisons for all the immune parameters among the four experimental groups were done by one-way ANOVA followed by Tukey’s post-test for multiple comparisons (α = 0.05). Analysis of the results between “protected” and “non protected” animals were all performed using the Mann–Whitney test. For correlation analyses, each pair of data sets were previously studied for normality using the Shapiro–Wilk test, and correlations were analyzed according to the Pearson’s correlation test and interpreted through the Pearson’s linear correlation coefficient. Finally, comparisons between related pairs of immune parameters within the same experimental group were done using a paired T-test. Statistical analyses were carried out using GraphPad Prism v5.0 (Prism, La Jolla, CA, USA).

## 3. Results

### 3.1. In Vivo Challenge with a Heterologous Strain of FMDV in Vaccinated Cattle

Four experimental groups of cattle were generated using different FMD vaccine formulations and immunization schedules. Experimental groups (*n* = 5 each) received either one dose of a lower antigen payload monovalent vaccine (10 µg of FMDV A24/Cruzeiro per dose), a higher payload monovalent vaccine (40 µg of FMDV A24/Cruzeiro per dose), or a trivalent formulation (containing 10 µg of each of the A24/Cruzeiro and C3/Indaial strains and 20 µg of the O1/Campos strain), or two doses of the lower payload A24/Cruzeiro monovalent vaccine at 0 and 15 days post-primary vaccination (dpv).

The effect of the vaccine formulation and the administration protocol in the protection against a heterologous infection with FMDV were tested at 30 dpv by an in vivo challenge assay performed using 10,000 infectious doses of virulent FMDV A/Arg/01. Two naïve animals were also included as a control group in the experiment (Table 1).

Five animals presented clinical signs compatible with FMD after heterologous challenge and were classified as non-protected: two (#84 and #104) from the group vaccinated with a single dose of 10 µg of FMDV A24/Cruzeiro (*A24 10 µg*), one (#105) from the group immunized with the higher payload monovalent formulation (*A24 40 µg*), and both naïve individuals (#329 and #377). Transient hyperthermia in the absence of other clinical signs, or lesions in the tongue at the injection site of the challenge were not considered for protective status. 

The time course of the clinical scores allowed distinguishing three groups of animals. Both naïve animals showed the highest clinical scores and exhibited values ≥ 4 from 3 days post-infection (dpi) until the end of the experiment. A second group consisted of three vaccinated animals, two from *A24 10 µg* group and one from the *A24 40 µg* group, presenting maximum clinical scores (between 2 and 3) at seven dpi and a delay in the progression of the disease compared to naïve cattle. Finally, a third group, comprising the remaining animals of the *A24 10 µg* group, were protected from the generalization of lesions and clinical disease but showed transient episodes of moderate hyperthermia at some point during the examination period. The rest of the animals did not show any sign of FMD during the week after the challenge and were not included in the chart (Figure 1).

All animals were subjected to the detection of antibodies against FMDV non-structural proteins (NSP). Antibodies against NSP were not found in some animals with generalized FMD, probably due to the early time post-infection testing (7 dpi). Virus isolation was then performed from whole blood samples (taken at 4 dpi) in all animals with clinical FMD and one protected steer per experimental group as negative controls. Positive virus isolation was only found in non-protected individuals (Table 1). Six out of the seventeen animals protected against the heterologous challenge were also positive for NSP antibodies, indicating that vaccination in these animals did not avoid virus replication, although they did prevent disease generalization. Only animals from the revaccinated group (*A24 10 µg × 2*) resulted in all cases negative for the anti-NSP antibody detection assay (Table 1).

### 3.2. Total and Neutralizing Anti-FMDV Antibodies Induced before the Heterologous Challenge

The humoral immune responses in the vaccinated cattle were initially assessed at 30 dpv using LPBE and VNT. Total FMDV-specific antibodies were measured by LPBE against all four viral strains included in this study. Antibody titers at 30 dpv exhibited significant differences among groups only for homologous responses against the A24/Cruzeiro strain. Only the experimental group immunized with the trivalent formulation (*A24/C3I/O1C*) showed significant differences from the other groups against the O1/Campos and C3/Indaial strains (Figure 2).

All experimental vaccines elicited robust homologous responses against the A24/Cruzeiro virus, showing mean LPBE titers between 20 and 180 times higher than those associated with the EPP_75_ for this viral strain [26]. Mean antibody titers of the *A24 10 µg × 2* group were significantly higher than those of the *A24 10 µg* and *A24/C3I/O1C* groups, while mean values of the steers immunized with the higher payload monovalent vaccine (*A24 40 µg*) were also higher than those of the *A24/C3I/O1C* group (Figure 2).

Although following a similar trend as the A24/Cruzeiro strain, mean titers against the A/Arg/01 strain were between 4 to 15 times lower than those against the homologous virus. In contrast to the homologous antibody responses, no statistical differences were found among groups (Figure 2). Despite their lower magnitude, mean anti-A/Arg/01 LPBE titers were all above the EPP_75_ titer calculated in this assay for the homologous challenge with this strain. This included the *A24 10 µg* group, which only reached 60% of protection to the heterologous challenge with A/Arg/01 strain in vivo (Table 1) but showed mean antibody titers 1.3 times above those of the EPP_75_ threshold for the same strain. 

The induction of FMDV-neutralizing antibodies in the vaccinated animals was assessed against the A24/Cruzeiro and A/Arg/01 strains, following a standard VNT assay recommended by the WOAH. As previously described for the LPBE, all experimental groups developed strong FMDV-specific responses against the A24/Cruzeiro strain, with mean VNT titers 15 to 30 times higher than those estimated for the EPP_75_ for this assay and viral strain. All groups receiving monovalent formulations elicited VNT titers higher than the trivalent formulation but showed no differences among them (Figure 3a).

Mean neutralizing titers against the A/Arg/01 strain in each vaccination group were 5 to 12 times lower than those against the A/24 Cruzeiro strain. As observed for the total FMDV-specific antibodies, in vivo results obtained using the A/Arg/01 strain as a heterologous challenge virus did not completely match VNT serological results or the EPP_75_ established for homologous infections with this strain (Figure 3b). Mean NAb titer in the *A24 10 µg* group, which presented only 60% of protection in vivo, was above the EPP_75_ value for A/Arg/01 in VNT, while the mean titers in the *A24/C3I/O1C* group, which showed 100% protection in vivo, were slightly below this threshold. Moreover, experimental groups performing differently in the in vivo tests showed no significant differences in VNT titers against the A/Arg/01 strain. Only revaccinated animals developed significantly higher mean VNT titers than the *A24/C3I/O1C* group, even though both groups presented 100% protection in vivo.

Seeking to confirm these observations, we compared the mean titers observed at 30 dpv in protected and non-protected steers for total and neutralizing antibodies. As shown in Figure 4, no differences were found between these two groups in none of these assays. Moreover, mean titers for protected and non-protected animals were approximately two times above the corresponding the EPP_75_ values calculated by LPB-ELISA and VNT and used to estimate homologous protection against the A/Arg/01 strain (Figure 4a,b).

Other comparisons performed to test the effect of the doses (*A24 10 µg* vs. *A24 10 µg × 2*), multivalency (*A24/C3I/O1C* group vs. *A24 10 µg* + *A24 40 µg* groups), and antigenic payload (*A24 10 µg* group vs. *A24/C3I/O1C* + *A24 40 µg* groups) did not reveal differences in mean total or neutralizing antibody titers against the challenge strain (Appendix A).

### 3.3. FMDV-Specific Cellular Immune Responses Induced in Each Experimental Group

The presence of antigen-specific memory T-cells in vaccinated and naïve animals was evaluated one day before the heterologous challenge, by measuring the ex-vivo production of IFN-γ from peripheral whole blood samples stimulated with inactivated FMDV A24/Cruzeiro and A/Arg/01 strains. Only vaccinated animals were able to produce in vitro antigen-specific IFN-γ; however, no statistical differences were found among experimental groups irrespectively of the stimulating strain. Likewise, no differences were found when comparing protected vs. non-protected steers, and mean IFN-γ production was similar in samples stimulated with either A24/Cruzeiro or A/Arg/01 strains (Appendix A).

### 3.4. Isotype Profiles of the Anti-FMDV Antibodies Induced before the Heterologous Challenge

The isotype composition of the FMDV-specific antibody responses against the A24/Cruzeiro and A/Arg/01 strains was analyzed to study other qualitative aspects of the humoral responses at 30 dpv.

IgM responses against the homologous strain were high at 30 dpv with log_10_ titers between 2.85 and 4.15, without statistical differences among groups. As previously mentioned for the LPBE and VNT assays, mean IgM responses against A/Arg/01 were lower than the homologous response, showing a higher dispersion compared to those against the A24/Cruzeiro strain (Appendix A). Similarly, mean IgM titers were equivalent between protected and non-protected animals (Appendix A). Interestingly, the only unprotected animal (#105) from those vaccinated with the higher payload monovalent vaccine (40 µg of FMDV A24/Cruzeiro per dose) showed the highest FMDV-specific IgM titer compared to the rest of the group.

Mean titers of the A24/Cruzeiro FMDV-specific IgG1 and IgG2 isotypes resembled those of the LPBE. All vaccines induced high mean antibody titers (log_10_ 4.0 to 5.0), presenting average values in the *A24 10 µg ×2* and *A24 40 µg* groups significantly above those of the *A24/C3I/O1C* group (Figure 5a,c). For both IgG isotypes, mean antibody titers against the A/Arg/01 strain dropped 20 to 40 times, and no significant differences were found among groups (Figure 5b,d).

No statistical differences were observed for protected vs. non-protected steers in IgG isotypes titers (Figure 5e,f). Nevertheless, mean anti-FMDV IgG2 titers were 3.5 times higher in the protected steers compared to those of the non-protected animals (Figure 5f), a tendency also exhibited in the IgG1/IgG2 ratio comparison (Figure 5g).

Similarly, no significant differences were found when comparing the number of doses or virus strain composition in the vaccines (data not shown). However, the effect of the antigenic payload was visible in the mean antigen-specific IgG2 titers, which were significantly higher for the highest vaccine payloads (40 μg, Figure 5i).

### 3.5. Avidity of the Anti-FMDV Antibodies Induced before the Heterologous Challenge

The avidity of the FMDV-specific humoral responses was assessed for IgG1 and IgG2 isotypes, and whole IgG antibodies against the A/Arg/01, A24/Cruzeiro, and O1/Campos strains. No statistical differences were found among experimental groups for these comparisons (Appendix A). Contrarily to those observed for the rest of the humoral parameters, avidity responses within each experimental group were similar between the A24/Cruzeiro and A/Arg/01 strains (Appendix A), even though total antibody titers corresponding to the A24/Cruzeiro strain were up to 40 times higher than those of the A/Arg/01 strain (Figure 2).

However, the evaluation of the individual IgG AI values against the heterologous challenge strain demonstrated that all three non-protected animals from the experimental infection presented the lowest AI among all vaccinated steers (Figure 6a). Consequently, the mean IgG AI against the A/Arg/01 strain in protected cattle was significantly higher than those in non-protected animals (Figure 6b).

As shown in Figure 7, the values of AI were similar between the A24/Cruzeiro and A/Arg/01 strains within each experimental group (see also Appendix A), although total antibody titers of the A24/Cruzeiro strain were much higher than those of the A/Arg/01 virus (Figure 2).

The only exception was the group of animals receiving the trivalent formulation (*A24/C3I/O1C*). These animals exhibited mean AI significantly higher for the A/Arg/01 strain than for the A24/Cruzeiro strain. This pattern was observed for total and isotype IgG responses, even when antibody titers against the A/Arg/01 strain were between 4 and 12 times lower than against the homologous virus (Figure 7).

Interestingly, comparisons for the A24/Cruzeiro and O1/Campos strains also showed that mean AI in protected steers were significantly higher than in non-protected animals (Figure 8).

Furthermore, correlation studies between the AI registered for each animal against the A24/Cruzeiro, A/Arg/01, and O1/Campos strains demonstrated strong positive correlations for all the tested FMDV strains (Figure 9). These results might suggest that high-affinity antibodies induced in vaccinated animals, irrespectively of the vaccine protocol, recognize a significant proportion of epitopes shared between these three strains. 

## 4. Discussion

Several papers have evaluated the in vivo cross-protection between field and vaccine strains of FMDV during the last two decades. These experiments were completed in different natural hosts, in some cases identifying vaccine formulation features that may improve cross-protection but without further details on the immune mechanisms potentially involved [11,24,31,32,33,34,35,36,37]. Other authors studied structural elements relevant for neutralization within the viral capsid, including those potentially implicated in cross neutralization [38,39,40]. The current study analyzed different vaccine protocols for their ability to prevent generalized FMD in cattle experimentally infected with a heterologous strain of FMDV, focusing on the host’s immune response.

Our work utilized a well-known model of heterologous protection between the FMD vaccine prototypical strain A24/Cruzeiro and the A/Arg/01 strain [41], isolated during the 2001 outbreaks in Argentina [24]. Goris et al., showed that cattle immunized with a single dose of a 10 μg monovalent FMDV A24/Cruzeiro vaccine, equivalent to the lower dose monovalent vaccine used in this study, presented 88.5% of homologous protection measured by PPG and percentages ranging from 56.3 to 12.5% against experimental infection with FMDV A/Arg/01 [41]. Our results demonstrate that the increment in the antigenic payload, the revaccination, or the addition of virus strains in the formulation, even from different serotypes to the challenge strain, may improve the performance of a high-potency monovalent FMD vaccine against infection with a heterologous virus.

In this study, all vaccine protocols elicited solid antibody responses against the A24/Cruzeiro strain, as measured by LPBE and VNT, two serological assays commonly used to evaluate homologous and heterologous protection [1]. Vaccine-induced antibody titers in the experimental groups showed values between 20 and 170 times above the EPP_75_ for LPB-ELISA and 10 to 30 times over the EPP_75_ for VNT.

Total and neutralizing antibody titers against the A/Arg/01 strain were lower than those against the A24/Cruzeiro strain. None of these serological parameters showed significant differences among experimental groups in most comparisons and did not match the in vivo heterologous challenge results. Discrepancies between the in vivo challenge and the LPBE and VNT results for the A/Arg/01 strain were also observed when referred to the corresponding EPP_75_ values: mean FMDV-specific neutralizing and total antibody titers in the *A24 10 µg* group (60% of protection in vivo) were higher than both EPP_75_ reference values, while the mean VNT titer in the *A24/C3I/O1C* group, which showed 100% of protection in vivo, was lower than the EPP_75_ threshold.

Since the EPP_75_ values represent reliable tools to predict homologous protection at a population level [23], these inconsistencies may be related to the limited number of individuals per experimental group (*n* = 5) but also to constraints in using this parameter for estimating heterologous protection (i.e., when the challenge virus and the EPP_75_ correspond to a strain not included in the vaccines). Moreover, when incorporating all vaccinated animals (*n* = 20) in the statistical assessments, no association was found between the clinical outcome of the experimental challenge and statistical differences in the mean titers measured by LPBE or VNT. These findings would indicate that considering only quantitative parameters, such as the total of neutralizing antibody titers, may result insufficient for assessing heterologous protection with FMDV.

Other parameters of the adaptive responses, such as the induction of memory T-cell responses or titers of antigen-specific IgM or IgG1 at 30 dpv, did not show a correlation to in vivo results, nor when comparing experimental groups, protected vs. non-protected animals, antigenic payload, or number of doses nor strains in the vaccine. 

Observations regarding the antigen-specific IgG2 responses were different. As seen for other humoral parameters, IgG2 antibody titers specific for A/Arg/01 strain were much lower (up to thirty times) than against the A24/Cruzeiro strain, although following similar response patterns. Despite this, mean IgG2 titers against the A/Arg/01 strain were 3.5 times higher in protected cattle vs. non-protected animals, an observation also reflected in the IgG1/IgG2 ratio comparison. Following this trend, mean IgG2 titers were significantly above for the highest payload vaccine groups (*A24/C3I/O1C* and *A24 40 µg*) than for animals in the *A24 10 µg* group. Previous reports from our group showed that after infection or vaccination in naïve cattle, FMDV-specific IgG2 and IgG2 antibody-secreting cells are found at lower levels and a few days after the detection of IgG1 [15,42]. Our results would indicate that the increase in the antigen payload may promote a faster maturation of the FMDV-specific B-cells, reflected in an earlier isotype switch. In this regard, it is interesting to note that the only animal in the higher payload monovalent vaccine group (*A24 40 µg*), which was not protected from the heterologous challenge (#105), also showed the highest FMDV-specific IgM and the lowest AI, IgG1, and IgG2 titers among its group at the time of the challenge, indicating a delayed development of the adaptive response against the virus.

The study of the avidity of the immune sera before the challenge also shed light on the immune processes involved in the heterologous protection. Mean AI was not different among experimental groups, exhibiting similar values when measured against the A/Arg/01 and A24/Cruzeiro strains. However, the avidity assay showed that all non-protected steers (#84, #104, and #105) presented the lowest AI values against the challenge strain. Consequently, as opposed to that observed with titers of total or neutralizing antibodies, mean AI in non-protected animals were significantly lower than in protected cattle. These results concur with others previously published for cattle immunized with A24/Cruzeiro vaccines and challenged with the A/Arg/01 strain [29]. A recent paper by Gordon et al. also demonstrates that blocking the association of the FMDV particles to dendritic follicular cells in mice does not decrease the FMDV-specific IgM and IgG titers but promotes a defective maturation of the affinity of the antibodies, which in turn reduces the avidity and the virus-neutralizing activity of the sera [43]. Thus, the maturation of the B-cell responses, reflected by the increase in the antigen-specific affinity and the resulting avidity of the sera, seems to play a critical role in the development of protective responses.

Interestingly, significant differences were also detected for the A24/Cruzeiro and the O1/Campos strains. Further analyses also proved a high correlation (Pearson’s r values between 0.91 and 0.95) among the AI values measured for each animal against the A/Arg/01, A/Cruzeiro, and O1/Campos strains. These results would suggest that independently of the number of high-affinity antibodies induced in the vaccinated animal and the immunization protocol received, the epitopes recognized by these high-affinity immunoglobulins may be coincident among these three strains, even though they belong to different serotypes.

Only two experimental groups presented 100% protection against the heterologous FMDV challenge: those immunized with the trivalent formulation (*A24/C3I/O1C*) and the revaccinated animals (*A24 10 µg × 2*). In contrast, a single dose of this lower dose monovalent vaccine only yielded 60% protection. This observation concurs with previous results, showing that revaccination with formulations containing FMDV A24/Cruzeiro improved their performance against heterologous FMDV strains from the same [24] or different serotypes [33]. Most probably, revaccination promotes the development of anamnestic responses, improving the quantity and quality of the adaptive response against the challenge strain. However, besides its performance in the heterologous challenge experiment and the observation that only the *A24 10 µg × 2* group was 100% negative for FMDV NSP detection at seven dpi, most of the parameters tested for the A/Arg/01 strain showed no differences between the revaccinated cattle and the rest of the experimental groups. Such results may be related to the revaccination and sampling time points selected for this experiment.

The group of animals immunized with the trivalent formulation was the other one showing 100% protection against the heterologous challenge. Other authors have also indicated an improved efficacy of multivalent vaccines compared to monovalent formulations in cattle using SAT strains [42]. Only those animals that received the trivalent vaccine showed significantly higher mean AI values against the A/Arg/01 strain than those against the A24/Cruzeiro strain. The preliminary analyses of vaccine-induced antibodies showing cross-reactivity between C3/Indaial, or O1/Campos and A/Arg/01 strains showed that depletion with C3I/Indaial or O1/Campos FMDV particles produced an average reduction of 50% in the antibody titers against the A/Arg/01 strain in animals with the trivalent formulation. Serum samples from animals immunized with monovalent formulations only generated reductions between 10% and 25% (Appendix A and Appendix B). These results indicate that the proportion of antibodies with diverse specificity induced in the trivalent formulation was higher than in monovalent vaccines; consequently, antibodies generated against the C- and O-serotype strains may have also contributed to the protective response registered in these animals.

Other authors have proposed different hypotheses to explain the enhanced protection observed for some multivalent immunogens and its relation to the affinity of the induced antibodies. An early work by Steward et al. associated protection against FMDV challenge in cattle to the induction of high-affinity antibodies against a synthetic peptide FMDV vaccine [44]. A study in mice immunized against a foreign protein (egg-white lysozyme) analyzed the patterns of specificity and avidity of the induced immunoglobulins by obtaining monoclonal antibodies generated after several immunizations. These authors propose that the affinity maturation observed in serum antibodies reflects enhances in the number and diversity of specific immunoglobulins rather than an increase in the average avidity of the antibody population, starting at seven days post-immunization [45]. In this same area, Chaudhury et al. presented a theoretical model of B cell affinity maturation that predicts the specificity and cross-reactivity of the antibody response in a malaria model during monovalent and polyvalent vaccinations [46]. Their results showed that immunization with multiple antigens, as opposed to single antigen vaccines, increased the proportion of B cells that bind to conserved epitopes, favoring the development of more reactive antibodies to variable epitopes absent in the vaccine. Such a mechanism, called “strain dilution” postulates that the polyclonal response of all antibodies together, following immunization with formulations comprising various antigens, may result in a protective response to multiple strains. Some of these conclusions coincide and may explain the results observed in this paper.

This work shows that besides other aspects of the adaptive responses promoted by vaccination not included in the study, such as activation of memory responses after infection [15,42,47,48,49], qualitative features of the humoral response represent significant factors in the protection against heterologous FMDV infection. These mainly include the diversity and maturation of the antibodies induced, which is reflected in the avidity and isotype profiles of the vaccine-induced sera. Moreover, the strong correlation among avidity values observed against FMDV strains from three distinct serotypes would suggest that high-affinity antibodies induced by vaccination may recognize similar epitopes within the capsid of the different FMDV serotypes. Thus, the induction of such high-affinity immunoglobulins against potentially invariant epitopes may also be relevant in generating cross-reactive antibodies, enabling further protection against a heterologous FMDV infection. Additional studies should be performed to assess and extend these findings to other FMDV strains and serotypes.

## Figures and Tables

**Figure 1 viruses-14-01781-f001:**
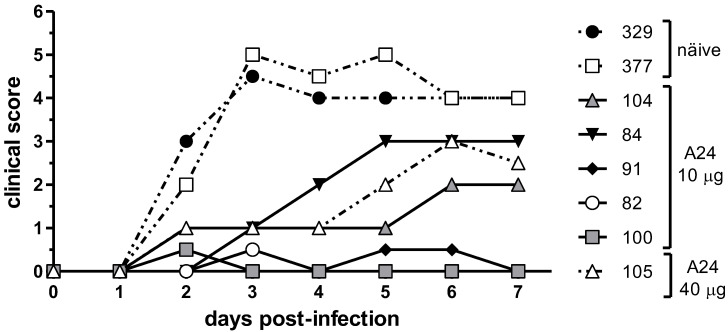
Progression of foot-and-mouth disease (FMD) clinical signs after heterologous challenge. Each line depicts the clinical scores registered for individual animals after infection with FMD virus (FMDV) A/Arg/01. The score was determined by assigning 1 point for hyperthermia > 40.0 °C and 1 point for each limb with vesicular lesions, reaching a maximum daily score of 5 points. Mild hyperthermia (≥39.5 °C and ≤40.0 °C) in the absence of other FMD-related signs was scored as 0.5 points.

**Figure 2 viruses-14-01781-f002:**
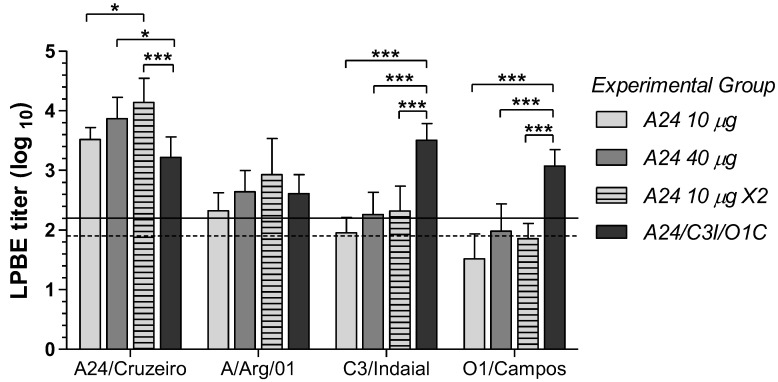
Total FMDV-specific antibodies induced in each experimental group at 30 days post-primary vaccination (dpv). Total anti-FMDV antibody titers measured by liquid-phase blocking ELISA (LPBE) against the A24/Cruzeiro, A/Arg/01, C3/Indaial, and O1/Campos FMDV strains. Results are expressed as the log10 of the mean LPBE titer in each experimental group at 30 dpv before heterologous challenge. Bars represent mean values of each experimental group ± SD. Dotted and black lines depict the titers corresponding to the expected percentage of protection ≥ 75% (EPP_75_) for A24/Cruzeiro (1.90) and A/Arg/01 (2.20) strains, respectively. Asterisks denote statistical differences between experimental groups (ANOVA) * *p* ≤ 0.05; *** *p* ≤ 0.001.

**Figure 3 viruses-14-01781-f003:**
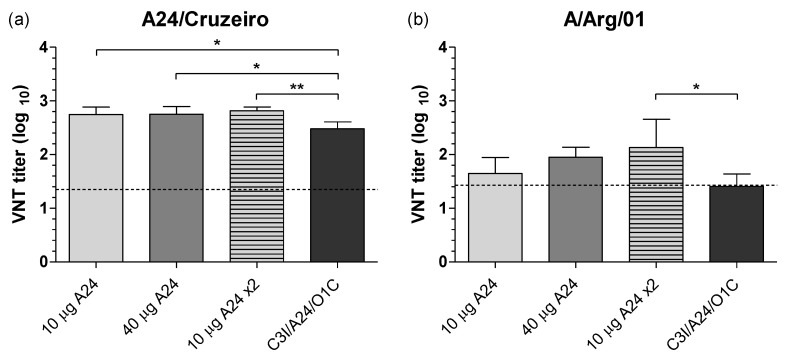
Neutralizing antibodies against FMDV A24/Cruzeiro and A/Arg/01 strains at 30 dpv. Serum neutralizing antibodies against FMDV A24/Cruzeiro (**a**) and A/Arg/01 (**b**) strains were measured by virus neutralization test (VNT) for all vaccinated animals. Results represent the mean of four independent experiments and are expressed as the log_10_ of the resulting mean titer in each experimental group at 30 dpv. Bars represent the mean value of each experimental group ± SD. Dotted lines depict VNT titers corresponding to the EPP_75_ for A24/Cruzeiro strain (1.35) and A/Arg/01 strain (1.43) in the corresponding charts. Asterisks denote statistical differences between experimental groups (ANOVA) * *p* ≤ 0.05 and ** *p* ≤ 0.01.

**Figure 4 viruses-14-01781-f004:**
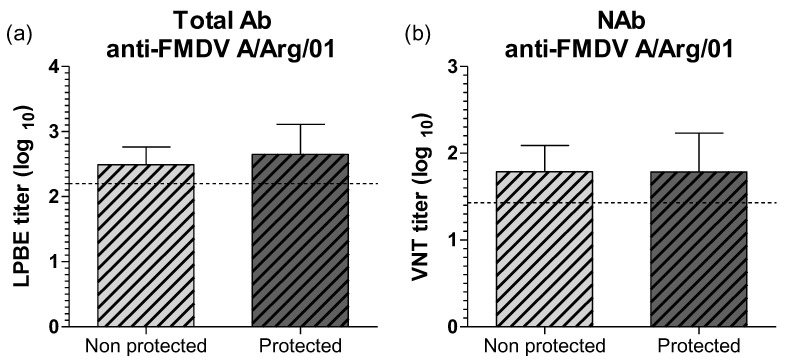
Total and neutralizing antibody responses against FMDV A/Arg/01 strain in protected and non-protected animals against the heterologous challenge at 30 dpv. Total FMDV-specific antibodies were measure by LPBE (**a**) and neutralizing antibodies were measure by VNT (**b**). Results are expressed as the log10 of the mean titers against heterologous strain, A/Arg/01. Dotted lines depict the titers corresponding to the EPP_75_ for A/Arg/01 strain for each serological assay. Bars represent mean value ± SD.

**Figure 5 viruses-14-01781-f005:**
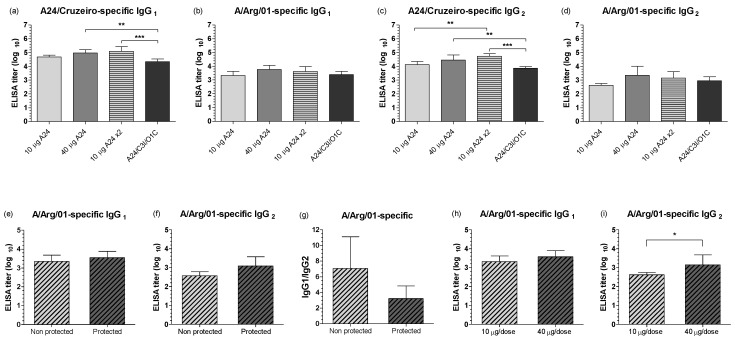
FMDV-specific IgG isotype responses in vaccinated animals at 30 dpv. (**a**–**d**) FMDV-specific IgG1 and IgG2 measured by ELISA for each experimental group against A24/Cruzeiro (**a**,**c**) and A/Arg/01 (**b**,**d**) strains. Results are expressed as the log_10_ of the mean titer for each vaccine group at 30 dpv. (**e**,**f**) IgG isotype responses in animals protected and non-protected to the heterologous challenge. Results are expressed as the log_10_ of the mean titers for IgG1 (**e**) or IgG2 (**f**) against the A/Arg/01 strain. (**g**) Ratio between mean IgG1 and IgG2 titers obtained by ELISA for protected and non-protected steers against the A/Arg/01 strain. (**h**,**i**) IgG isotype responses against FMDV A/Arg/01 strain according to the antigenic payload of the formulations. Animals were grouped as 10 μg/dose (including the *A24 10 µg* group) or 40 μg/dose (including the *A24/C3I/O1C* and *A24 40 µg* groups). FMDV-specific IgG1 (**h**), and IgG2 (**i**) antibodies against the A/Arg/01 strain at 30 dpv were measured by ELISA, expressed as the log_10_ of the mean titers. For all panels, bars represent mean values of each group ± SD. Asterisks denote statistical differences between experimental groups * *p* ≤ 0.05 (Mann–Whitney test) and ** *p* ≤ 0.01 *** *p* ≤ 0.001 (ANOVA).

**Figure 6 viruses-14-01781-f006:**
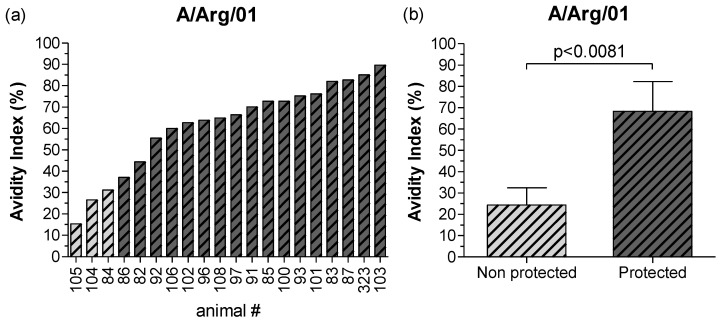
Avidity of the IgG antibodies against the FMDV A/Arg/01 at 30 dpv. The avidity of the FMDV-specific IgG antibody responses against the A/Arg/01 strain was measured by Avidity ELISA. (**a**) Avidity indexes (AI) for all vaccinated animals following an increasing order. Non-protected animals are indicated as clear shaded bars. (**b**) Mean AI in animals protected and non-protected against heterologous challenge at 30 dpv. Bars represent the mean value ± SD. Statistical differences were determined by Mann–Whitney test.

**Figure 7 viruses-14-01781-f007:**
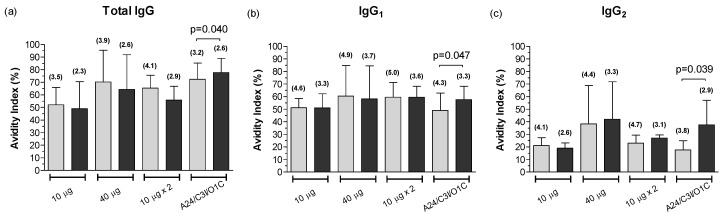
Mean avidity of the FMDV-specific IgG antibodies against the A24/Cruzeiro and A/Arg/01 strains at 30 dpv. The avidity for FMDV-specific whole IgG (**a**), IgG1 (**b**), and IgG2 (**c**) antibodies were measured by avidity ELISA against A24/Cruzeiro (light bars) and A/Arg/01 strains (dark bars) for experimental group. Results are expressed as the AI and bars represent the mean AI ± SD. Values in parentheses represent the corresponding ELISA mean antibody titers expressed as log_10_ for each experimental group. Statistical differences were determined by a paired *t*-test.

**Figure 8 viruses-14-01781-f008:**
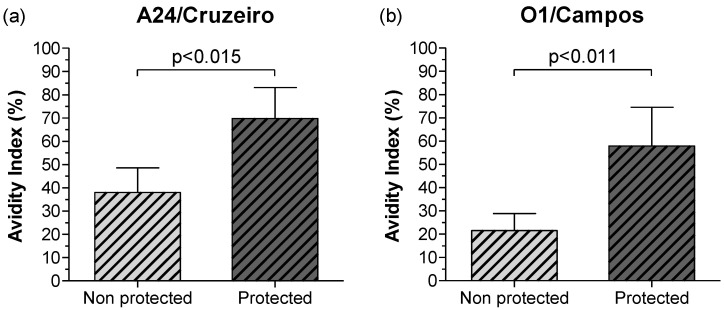
Mean avidity of the FMDV-specific total IgG antibodies against the A24/Cruzeiro and O1/Campos strains at 30 dpv. Animals were grouped according to the results in the in vivo FMDV A/Arg/01 challenge at 30 dpv. Mean AI of the FMDV-specific IgG antibody responses against the A24/Cruzeiro (**a**) and O1/Campos (**b**) strains were measured by Avidity ELISA. Results are expressed as the mean AI and bars represent the mean value ± SD. Significant statistical differences between protected and non-protected steers are indicated by the corresponding *p* values (Mann–Whitney test).

**Figure 9 viruses-14-01781-f009:**
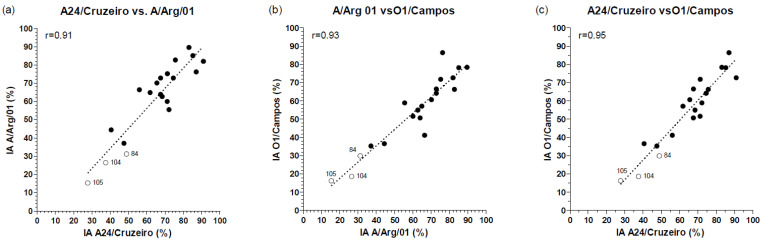
Correlation of avidity indexes against different FMDV strains in serum samples from vaccinated animals at 30 dpv. The AI of the FMDV-specific IgG antibodies measured against the A24/Cruzeiro, A/Arg/01 and O1/Campos strains in each vaccinated animal were analyzed by pairs using a Pearson’s correlation test, as indicated in each panel. White circles denote non-protected animals (#84, #104, #195), and the corresponding Pearson’s *r* values are shown in each graph (*p* value < 0.0001).

**Table 1 viruses-14-01781-t001:** Percentage of protection and viral replication in FMD vaccinated animals after heterologous challenge. The performance in the in vivo challenge assay is expressed as the percentage of protected animals in each experimental group. Rows represent individual animals; shaded rows correspond to non-protected animals. Viral replication after experimental infection was assessed from serum samples taken at 7 dpi through the detection of antibodies against FMDV non-structural proteins (NSP). Results express the presence (+) or absence (-) of anti-FMDV NSP antibodies. FMDV isolation (VI) was performed in serum samples obtained at 4 dpi from animals with clinical FMD. The development (+) or the absence (-) of cytopathic effect was assayed on BHK-21 cell cultures (ND: non-determined).

Experimental Group	% of In Vivo Protection	Animal #	In Vivo Protection	NSP 7 dpi	VI 4 dpi
*A24 10 µg*	60%	82	yes	-	ND
		84	no	-	+
		91	yes	+	-
		100	yes	+	ND
		104	no	+	+
*A24 40 µg*	80%	83	yes	-	ND
		85	yes	+	ND
		93	yes	-	-
		101	yes	-	ND
		105	no	-	+
*A24 10 µg × 2*	100%	86	yes	-	ND
		92	yes	-	ND
		96	yes	-	ND
		102	yes	-	-
		106	yes	-	ND
*A24/C3I/O1C*	100%	87	yes	+	ND
		97	yes	-	-
		103	yes	-	ND
		108	yes	+	ND
		323	yes	+	ND
*naïve*	0%	329	no	+	+
		377	no	+	+

## Data Availability

Not applicable.

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
