# Peer review of "Assessment on Different Vaccine Formulation Parameters in the Protection against Heterologous Challenge with FMDV in Cattle"

_viruses, 2022, doi:10.3390/v14081781_

Round 1

Reviewer 1 Report

Giacomo et al. used different vaccine formulations in this study and analyzed their parameters after the heterologous FMDV challenge. They showed that revaccination, dose, and multivalency might affect the clinical outcome post-FMDV heterologous challenge. Furthermore, the relationship between the protection from the virus challenge and qualitative traits of the induced antibodies was explained. The study design is appropriate, all the experiments were conducted properly, and the results were presented elegantly. Some minor comments are given below.

Line 112 Assessment of FMDV-Specific Total and Neutralizing Antibodies, the heading should be like this.

In Line number 321, the naïve animal number was described as 329, but in table 1, it is described as 377; please make it consistent.

Line 627, replace viruses with FMDV serotypes. It may confuse the readers.

Author Response

We thank the reviewer for helpful suggestions and thorough reading. We have included the following modifications in the revised version:

-  The heading in section 2.5 has been modified as indicated (line 112)

-  The numbers of the naïve individuals have been corrected to be consistent with Table 1 (line 230)

-  The word “viruses” has been replaced by “FMDV serotypes” as suggested (line 643)

Reviewer 2 Report

July 17th, 2022

Review: viruses-1829449

“Assessment on Different Vaccine Formulation Parameters in the Protection against Heterologous Challenge with FMDV in Cattle”

In this manuscript the authors describe the analysis of different aspect of mainly humoral but also some cellular immunoresponse to vaccination against foot-and-mouth disease aiming to recognize correlates of protection when challenge with heterologous viruses. The topic is of great interest because frequently during outbreaks, there is not time for vaccine matching, and vaccination campaigns to control FMD use vaccines available, even being formulated with different serotypes from the circulating one. The phenomenon of vaccine crossprotection has being under research for FMD for decades and still the parameter behind protective immunity are poorly understood. In this very well written manuscript, the authors described the design of several groups under different vaccination conditions to ensure different levels of crossprotection. However, in our opinion, the authors failed to include an important control of animal immunized with homologous A/Arg/01. First of all, this would have corroborated that challenge with this virus strain in homologous immunized animals can conferred 100%, and also, parameters of immunoresponse could have been used as reference for further analysis. How immunogenic is A/Arg/01 with respect to A24/Cru?

Minor comments:

1.     Clinical evaluation of the animals includes high temperatures and that can be confusing, because high temperature that never progresses to appearance of vesicles, is it considered generalization? Should animals 91, 92 and 100 be considered unprotected when they never showed FMD vesicles?

2.     It is not clear how do the authors evaluate the production of IFNgamma after ex vivo stimulation. Do they use ELISpot, ELISA of PBMCs supernatants or flow cytometry. Please, clarify in the methods.

Author Response

The reviewer raises an important point. We relied on earlier works to know the immunogenic and protective capacities of vaccines formulated with the A/Arg/01 strain against infections with the same strain. For this purpose, we used the same serological assays described in these papers. In Mattion et al., a monovalent FMD vaccine containing 10 μg/dose of the A/Arg/01 strain provided 80% of protection at 30 dpv, and a polyvalent vaccine containing 6 μg/dose of FMDV A/Arg/01 conferred 81% of protection at 90 dpv, measured by the PGP method against this strain (1). Moreover, these authors showed how the inclusion of the A/Arg/01 strain in the national vaccination campaigns reduced and controlled the FMD field outbreaks due to this strain in Argentina. A few years later, Maradei et al. presented a correlation between the LPBE antibody titers against the FMDV A/Arg/01 (n=177) and PGP results obtained in 11 challenge experiments (PGP, 10 x 16 animals + 1 x 17 animals= 177) with the same strain (2). These correlation studies established an expected percentage of protection (EPP) against the A/Arg/01 challenge according to the strain-specific antibody titers induced in cattle immunized with vaccines containing this FMDV strain. This method, as described in section 2.6 of our work (lines 121-131), is currently used by the sanitary authorities in Argentina to evaluate FMD vaccine potency: the EPP ≥75% (EPP75) values serve as a reference for the strain-specific antibody titers (measured by LPBE or VNT) associated with the protection (at the population level) against the homologous challenge with A/Arg/01 strain. Both reports demonstrated that the A/Arg/01 strain is immunogenic and may provide protection against infection with the homologous strain when included in inactivated FMDV oil vaccines. As is shown in Maradei et al. (2), the A/Arg/01 strain elicited similar levels of antigen-specific antibodies compared to other virus vaccine strains, providing equivalent levels of protection against the homologous challenge. We used the EPP75 as reference values for protective antibody titers induced by the experimental vaccines. However, although the EPP75 is an excellent tool for controlling vaccines against infection with homologous strains (i.e., included in the formulations), we showed that it may result less accurate when evaluating protection against infections with heterologous strains (i.e., not included in the formulations). Besides considerations around the number of individuals per group, we interpreted that the LPBE and the VNT assays mostly consider quantitative parameters, such as total o neutralizing antibody titers, and may result insufficient for assessing heterologous protection with FMDV. Both references are included in the corresponding section.

(1) Mattion, N. et al. Reintroduction of Foot-and-Mouth Disease in Argentina: Characterisation of the Isolates and Development of Tools for the Control and Eradication of the Disease. Vaccine 2004, 22, 4149–4162, doi:10.1016/j.vaccine.2004.06.040.

(2) Maradei, E. et al. Updating of the Correlation between LpELISA Titers and Protection from Virus Challenge for the Assessment of the Potency of Polyvalent Aphtovirus Vaccines in Argentina. Vaccine 2008, 26, 6577–6586, doi:10.1016/j.vaccine.2008.09.033.

Other comments:

  1. Transient and moderate hyperthermia that never progresses to the appearance of vesicles was not considered FMD generalization. Animals #82, #91 and #100 only suffered transient and moderate hyperthermia (between 39.5°C and 40°C) without developing epithelial vesicles and were considered “protected”. This information has been clarified in section 2.3 of the revised version (lines 105-106).
  2. FMDV-specific IFN-gamma production was assessed by a commercial ELISA using plasma from whole blood samples stimulated in vitro with purified inactivated FMDV particles from the A24/cruzeiro and A/Arg/01 strains. A more detailed protocol has been included in the revised version of the manuscript (lines 162-174).